# Interrogating the Metabolomic Profile of Amyotrophic Lateral Sclerosis in the Post-Mortem Human Brain by Infrared Matrix-Assisted Laser Desorption Electrospray Ionization (IR-MALDESI) Mass Spectrometry Imaging (MSI)

**DOI:** 10.3390/metabo12111096

**Published:** 2022-11-10

**Authors:** Alexandria L. Sohn, Lingyan Ping, Jonathan D. Glass, Nicholas T. Seyfried, Emily C. Hector, David C. Muddiman

**Affiliations:** 1FTMS Laboratory for Human Health Research, Department of Chemistry, North Carolina State University, Raleigh, NC 27695, USA; 2Goizueta Alzheimer’s Disease Research Center, Emory University School of Medicine, Atlanta, GA 30322, USA; 3Department of Neurology, Emory University School of Medicine, Atlanta, GA 30322, USA; 4Department of Biochemistry, Emory University School of Medicine, Atlanta, GA 30322, USA; 5Department of Statistics, North Carolina State University, Raleigh, NC 27695, USA; 6Molecular Education, Technology and Research Innovation Center (METRIC), North Carolina State University, Raleigh, NC 27695, USA

**Keywords:** IR-MALDESI MSI, mass spectrometry imaging, Amyotrophic lateral sclerosis (ALS), neurodegenerative disease, multiomic

## Abstract

Amyotrophic lateral sclerosis (ALS) is an idiopathic, fatal neurodegenerative disease characterized by progressive loss of motor function with an average survival time of 2–5 years after diagnosis. Due to the lack of signature biomarkers and heterogenous disease phenotypes, a definitive diagnosis of ALS can be challenging. Comprehensive investigation of this disease is imperative to discovering unique features to expedite the diagnostic process and improve diagnostic accuracy. Here, we present untargeted metabolomics by mass spectrometry imaging (MSI) for comparing sporadic ALS (sALS) and *C9orf72* positive (C9Pos) post-mortem frontal cortex human brain tissues against a control cohort. The spatial distribution and relative abundance of metabolites were measured by infrared matrix-assisted laser desorption electrospray ionization (IR-MALDESI) MSI for association to biological pathways. Proteomic studies on the same patients were completed via LC-MS/MS in a previous study, and results were integrated with imaging metabolomics results to enhance the breadth of molecular coverage. Utilizing METASPACE annotation platform and MSiPeakfinder, nearly 300 metabolites were identified across the sixteen samples, where 25 were identified as dysregulated between disease cohorts. The dysregulated metabolites were further examined for their relevance to alanine, aspartate, and glutamate metabolism, glutathione metabolism, and arginine and proline metabolism. The dysregulated pathways discussed are consistent with reports from other ALS studies. To our knowledge, this work is the first of its kind, reporting on the investigation of ALS post-mortem human brain tissue analyzed by multiomic MSI.

## 1. Introduction

Amyotrophic lateral sclerosis (ALS) is a fatal neurodegenerative disease [1,2,3,4] characterized by the degradation of upper and lower motoneurons and progressive loss of motor function, where patients diagnosed with ALS often pass from respiratory failure within 2–5 years following diagnosis [5,6]. The incidence rate of ALS is about 2 per 100,000 individuals and is primarily categorized based on familial inheritance or lack thereof [1,4]. In 10% of cases, patients have more than one occurrence within a family line and are considered having familial ALS (fALS) [2,7]. Conversely, 90% of individuals develop ALS at seemingly random incidence and are diagnosed with sporadic ALS (sALS) [8]. Mechanisms of sALS are not well understood, but hypotheses span from glutamate toxicity, dysfunctional RNA metabolism, mitochondrial dysfunction, oxidative stress, and beyond [2].

More than 20 genetic mutations have implications in both sALS and fALS, including *SOD1*, *TARDBP*, *FUS*, and Chromosome 9 open reading frame 72 (*C9orf72*). *C9orf72* is the most prevalent of these mutations, contributing to approximately 34% of fALS and 5% of sALS cases diagnosed [7]. Distinguished by numerous hexanucleotide repeat expansions (i.e., GGGGCC_n_), the mechanism of *C9orf72* pathogenesis is debated between loss-of-function, gain-of-function, or a combination of both [9]. The loss-of-function mutation is associated with haploinsufficiency of the *C9orf72* protein, potentially resulting in increased neuroinflammation [10]. RNA toxicity is the proposed gain-of-function mechanism of pathogenesis where 1) the sense and antisense transcripts are produced from the hexanucleotide repeat, promoting sequestration of essential RNA processing proteins [11], and/or 2) toxic dipeptide repeats (DPRs) are produced from non-AUG initiated translation [12,13] and can interact with ribosomal subunits or induce oxidative stress [14,15,16]. While these mechanisms have been proposed in relation to *C9orf72*, they are still debated and poorly understood [1].

Unlike other neurodegenerative diseases where clinical protocols are supplemented with medical testing [17,18,19], ALS diagnosis relies heavily on clinical procedures based on symptom presentation and progression [20,21,22]. Regardless of extensive testing and specialist referral, ALS is notoriously misdiagnosed with a delayed diagnostic timeline, commonly extending beyond twelve months [23,24]. Accurate diagnosis and monitoring of ALS urgently requires robust biomarkers and an enhanced understanding of its metabolic signature [1,25,26].

Considerable efforts to study ALS by mass spectrometry have been specific to proteomics and have provided insights into emerging protein biomarkers that may be indicative of disease progression [27,28]. While proteomics offers an important perspective about biological changes, integrating other omics data can provide additional context for a disease state. Metabolomics is another area of interest in biomarker discovery due to the inherent sensitivity of metabolomic phenotypes responding to continuous changes in a biological system [26,29,30]. Mass spectrometry imaging (MSI) is an attractive approach for analyzing metabolites in a biological sample [31,32], as it measures the abundances and spatial distributions of analytes across the sample without the necessity of labeling [33,34,35,36].

Infrared matrix-assisted laser desorption electrospray ionization (IR-MALDESI) MSI is an ambient ionization source offering distinct advantages such as soft ionization and high salt tolerance [36,37,38]. In IR-MALDESI analyses, an energy-absorbing ice layer is deposited on top of the sample prior to analysis; the mid-IR laser (2.97 µm) resonantly excites the O-H stretching bands of water to desorb neutral species from the sample, which are subsequently post-ionized in an electrospray plume [38]. Ice matrix is advantageous as it facilitates the desorption and ionization of analytes without introducing extra molecule interference [39,40]. Ultimately, these intrinsic characteristics of IR-MALDESI demonstrate its capacity for untargeted metabolomics and beyond [41,42,43,44,45,46].

Herein, we present untargeted metabolomic coverage via IR-MALDESI MSI of post-mortem human frontal cortex tissue in various ALS cohorts: control, sALS, and sALS *C9orf72* positive (C9Pos) cases. Previous proteomics measurements were pursued in the frontal cortex via LC-MS/MS in a previous proteomic study, and all MSI experiments reported here were completed on complimentary tissues from the same patients for accurate multiomic integration [47]. This investigation presents a multiomic perspective by highlighting metabolomic, MSI-specific discoveries and integrating proteomic data. The associated pathways discussed here may be potential avenues for biomarker discovery and enhancing our understanding of ALS.

## 2. Materials and Methods

### 2.1. Case Details and Sample Preparation

All post-mortem human brain tissues were provided by Emory Alzheimer’s Disease Research Center (ADRC) (Atlanta, GA, USA) and were acquired under proper compliance with Institutional Review Board (IRB) standards, as stated in a previous study. All patients were diagnosed and cared for by JDG. Cases selected for this study were from the same cohort previously published for proteomic analysis [47]. Subject cohorts included control (ctrl) (*n* = 5), sporadic ALS (als) (*n* = 6), and sporadic ALS with *C9orf72* mutations (c) (*n* = 5). *C9orf72* mutations were confirmed previously in blood samples by a primed polymerase chain reaction (PCR) method PMID 27488601) [47]. All patient case information is available in Appendix A. Frontal cortex brain tissues were mounted to a specimen disc with optimal cutting temperature (OCT) medium, sectioned at 10 µm, and thaw-mounted onto glass slides. The glass slides were packed in a slide container and shipped on dry ice from Emory University (Atlanta, GA, USA) to North Carolina State University (Raleigh, NC, USA) overnight. All samples were stored at −80 °C until time of analysis.

### 2.2. IR-MALDESI MSI

The IR-MALDESI MSI experimental design is summarized in Figure 1. Samples were placed on a Peltier-cooled translation stage cooled to −8 °C, enclosed in a humidity-controlled chamber, and purged with nitrogen gas (Arc3 Gases, Raleigh, NC, USA) to form a thin energy-absorbing ice layer. An electrospray ionization (ESI) plume comprised of 50% acetonitrile in water (Fisher Scientific, Hampton, NH, USA) and 0.2% formic acid (Sigma Aldrich, Carlsbad, CA, USA) was achieved by applying 3400V to the emitter tip and a solvent flow rate of 1.2 µL/min. A 2970 nm laser (JGMA, Burlington, MA, USA) was used to desorb the neutral species from the region of interest (ROI) across the sample. The laser energy applied was of 1.1 mJ/burst (10 pulses-per-burst) at a pulse rate of 10 kHz [48]. Tissue sections (*n* = 16) were completely randomized prior to analysis and imaged at a spatial resolution of 150 µm.

The IR-MALDESI source was coupled to an Exploris 240 (Thermo Fisher Scientific, Bremen, Germany) set at a resolving power of 240,000_FWHM_ at *m*/*z* 200. Metabolites were analyzed in positive polarity between *m*/*z* 75–400. The Automatic Gain Control (AGC) was disabled and a fixed injection time of 15 ms was utilized to synchronize laser ablation events and allow optimal ion accumulation within the ion routing multipole of the Exploris 240 [45]. The EASY-IC internal calibrant, fluoranthene (M●+ = *m*/*z* 202.0777), was enabled and the multi-injection RF threshold was set to 6.

The MATLAB-based IR-MALDESI MSI control software, RastirX, allows the users to define a ROI for MSI experiments. While rectangular ROIs are conventionally used, arbitrary (polygonal) ROIs were used to reduce analysis time, preserve sample integrity, and reduce downstream file size [49]. After analysis, the respective location file is produced by RastirX and is used for downstream file conversion.

### 2.3. IR-MALDESI MSI Data Analysis

RAW files were converted to mzML files using MSConvert [50,51], then subsequently converted to imzML files using the respective location file from arbitrary ROI sampling in imzML Converter [49,51]. All imzML files were uploaded to METASPACE [52] for putative identification based on spatial chaos, spectral isotope, and spatial isotope. Metabolite identifications were reported at a 10% false discovery rate using the HMDB-v4 database. MSiReader was used to generate ion images and MSI data analysis [53,54]. The MSiPeakfinder tool was used to compare two ROIs and isolate peaks that were either (1) present in at least 80% of the interrogated region and less than 20% of the reference region, or (2) present in greater than 20% of the reference region but 2× more abundant in the interrogated region; each cohort was treated as the interrogated and reference region to fulfill all possible combinations (i.e., sALS vs. C9Pos, sALS vs. control, C9Pos vs. sALS, C9Pos vs. control, control vs. sALS, control vs. C9Pos). Entire sample cohorts were used to distinguish disease-specific ions as opposed to evaluating biological variability. The ions reported by MSiPeakfinder were searched in the METLIN database with a 2.5 ppm mass tolerance, and their spectral accuracy was evaluated for putative identification.

To account for variability in analysis conditions and artifacts relevant to changes in background ions, all heat maps were normalized to the local TIC in MSiReader [55].

### 2.4. Metabolomic Pathway Analysis and Proteomic Integration

Variation in metabolite abundances was visualized in part by heat maps using Plotly [56]. To identify dysregulated pathways, pathway impact graphs for annotated metabolites were produced in MetaboAnalyst 5.0 with the Pathway Analysis Module [57].

To integrate the proteomics data, unique protein IDs (accession IDs from Proteome Discoverer) were extracted from the previous proteomic study [47] and were manually converted to their respective protein names using Universal Protein Resource (Uniprot) database [58]. Protein names were cross-searched in Kyoto Encyclopedia of Genes and Genomes (KEGG) to find metabolomic conversions [59]. Significant pathways were visualized in KEGG and ion images from MSI detection were incorporated to produce annotated pathways and metabolomic conversion figures.

### 2.5. Statistical Testing and Analysis

Volcano plots were constructed using *p*-values from Student’s *t*-tests for initial comparison of metabolite detection between cohorts, where a *p*-value (α) ≤ 0.05 was considered statistically significant. Cohorts were compared in pairs to fulfill all possible combinations (i.e., control vs. sALS, control vs. C9Pos, and sALS vs. C9Pos) to evaluate the groups’ metabolomic detection by both fold-change and statistical significance.

Metabolomic conversion figures show ion images with respective box plots of each metabolite, where each data point corresponds to the average ion abundance detected for each sample in the cohort. After using Shapiro-Wilk tests (*p* ≤ 0.05), each cohort for the metabolites reported were assumed to be normally distributed. Once the data were tested for Normality, one-way analysis of variance (ANOVA) tests were performed to compare the effect of disease cohort (i.e., control, sALS, C9Pos) on metabolite detection (e.g., tryptamine). If ANOVA testing yielded statistically significant results (*m*/*z* 130.0862 only), Tukey’s honest significant difference (HSD) post hoc testing was performed to evaluate statistical significance between individual cohorts in the study (e.g., control vs. sALS, control vs. C9Pos).

## 3. Results and Discussion

Multiomic data integration was accomplished in this work by combining proteomic coverage, completed via LC-MS/MS by the Seyfried group, and untargeted metabolomics via IR-MALDESI MSI utilizing high-resolution accurate mass (HRAM) mass spectrometry. By combining high-mass measurement accuracy (MMA) (±2.5 ppm), spectral accuracy (SA), and biological context as evidence, we assigned confident putative identifications based on MS1 data collected in imaging experiments, which are summarized in Appendix A. A total of 298 metabolites were identified and compared across the sample groups for analysis to evaluate differences in detection between cohorts.

### 3.1. Comparing Metabolomic Detection between ALS Cohorts

Heatmaps in Figure 2 were compared to discriminate any differentially expressed metabolites within an individual cohort in the study. Raw abundance values were *z*-transformed to visualize data relative to the mean and standard deviation of individual samples. Equation (1) shows the transformation of the ion abundances to the *z*-score (*Z*) used in the heat maps:(1)$$Z=\frac{x-\mu }{\sigma }$$
where *x* is the average ion abundance of an individual ion across all scans, *μ* is the mean abundance of all measured ions, and *σ* is the standard deviation of all measured ions, where all variables are calculated with respect to one sample of interest (e.g., als1, ctl1, c1). This visualization approach can be helpful in discerning consistent detection within a disease classification, indicating metabolites that may be characteristic to that respective group.

To view abundance differences closely, the full *m*/*z* window (*m*/*z* 75–400) was split into two smaller *m*/*z* windows, as displayed in Figure 2A,B. It was found that arachidonic acid (*m*/*z* 305.2475) and oleic acid (*m*/*z* 283.2631) generally present higher *z*-scores in Figure 2B but show some biological variation between samples, such as higher detection in patients ctl2 and ctl3 in the control group. This trend is consistent in the lower *m*/*z* range as well (Figure 2A), with piperidine (*m*/*z* 86.0964) and benzaldehyde (*m*/*z* 107.0491). Ion images of these analytes show their respective spatial distributions across each sample (Figure 2C). It was expected that the metabolite detection would differ from disease groups to the control as a result of motor function loss in sALS and C9Pos cases. However, Figure 2 shows consistent detection of analytes regardless of diagnosis. While this result doesn’t indicate drastic metabolic changes between the cohorts, this implies consistency and repeatability of IR-MALDESI, as analyses were conducted over several days [55].

Three volcano plots were created to highlight any potential metabolite dysregulation by comparing all combinations of disease cohorts, where fold changes were calculated as the ratio shown with each plot (e.g., Control:C9Pos) (Figure 3). Most metabolites in this study did not exceed the threshold of significance (*α* = 0.05) between disease cohorts, presenting only 25 metabolites either with a significant fold change or being statistically significant with an absolute fold change of two or more (summarized in Appendix A). Interestingly, most metabolites were downregulated in control and C9Pos cohorts relative to sALS, as shown in Figure 3B,C. Several metabolites indicated are amino acids, which is consistent with other ALS studies highlighting the disruption of amino acid metabolism [25,60,61]. This includes the neurotransmitter glutamate, which is continuously investigated for its role in oxidative stress and neuron cell death related to ALS [62,63].

### 3.2. Identification of Relevant Metabolic Pathways

To evaluate other potential mechanisms related to ALS, we next targeted relevant metabolic pathways of interest. Significantly different metabolites, as indicated by the volcano plots, were searched in MetaboAnalyst to visualize pathway dysregulation via pathway impact graphs. The size and intensity of red of the data points match and display the pathway influence and significance of the metabolites. Figure 4A highlights several pathways of interest for all metabolites identified, which are likely a function of an operating biological system. Alternatively, Figure 4B shows the pathways more prone to perturbation due to ALS by confining our “species of interest” only to those with some significance indicated by the volcano plots in Figure 3. These include the following annotated biological pathways: alanine, aspartate, and glutamate metabolism, glutathione metabolism, and arginine and proline metabolism (Figure 5, Figure 6 and Figure 7). Additionally, arginine biosynthesis was annotated but is not often discussed in literature, presenting a novel pathway that could be investigated in future studies (Appendix A).

### 3.3. Multiomic Integration

After identifying other metabolites involved in each pathway by utilizing the KEGG pathway database, metabolite detection by IR-MALDESI was validated as previously described and ion images were positioned accordingly with colored blocks separating the cohorts (e.g., C9Pos samples in pink). Proteins responsible for metabolite conversions were searched for in Umoh et al. data and are represented as circles along the pathway, where purple- and white-filled circles indicate detection or lack thereof, respectively [47]. Given glutamate’s prevalence in ALS literature, it was particularly interesting that we found this metabolite to be dysregulated. However, despite its detection in a variety of biological samples, glutamate has not been reported consistently as up- or down-regulated regarding ALS [25]. Evidence supports that glutamate is responsible for excitotoxicity in neurons and oxidative stress in mitochondria, ultimately promoting neuronal apoptosis that may be affiliated with ALS [61,63,64]. This neurotransmitter was detected by IR-MALDESI and is involved in crosstalk between several pathways. The pathway in Figure 5 shows the ion images of glutamate and other detected metabolites in alanine, aspartate, and glutamate metabolism, which was reported in other studies with potential implications for ALS pathophysiology [25,60,61].

Differences in spatial distribution are most apparent between asparagine and GABA, as an example (starred in Figure 5), where GABA is biased to certain regions of the tissue, and asparagine is sparsely detected across all patient samples. No obvious trends in metabolite up- or down-regulation are observed downstream between disease cohorts in this data, despite detecting glutamate dysregulation between the control and C9Pos classifications. However, this finding is consistent with proteomic results reported by the Seyfried group and other metabolomic investigations of ALS [47].

**Figure 5 metabolites-12-01096-f005:**
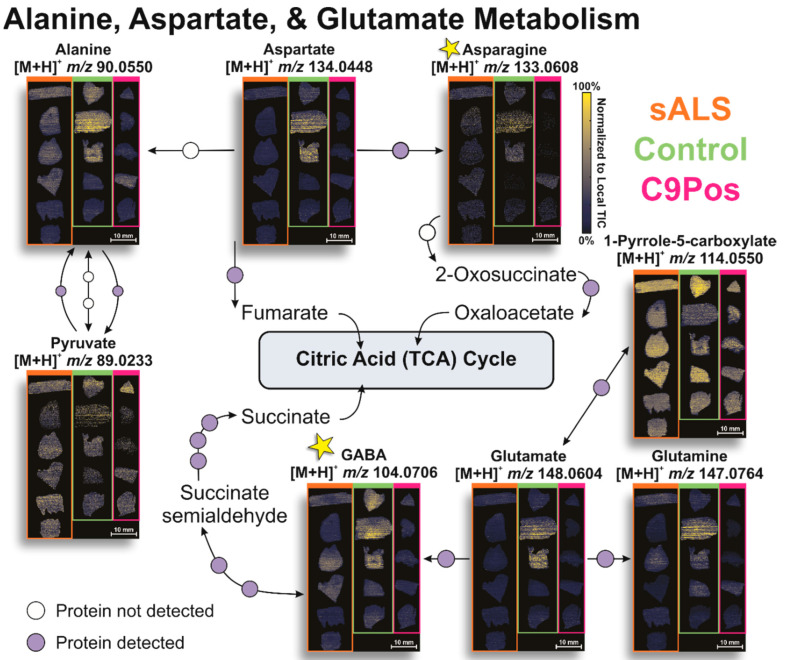
Annotated alanine, aspartate, and glutamate pathway with associated metabolite ion images by IR-MALDESI. Other metabolites associated with glutamate are detected in this study and are shown in ion images grouped with respect to disease classification (top right). Purple circles along conversion arrows indicate protein detection, while white circles are indicative of no detection in the post-mortem brain samples from previous proteomic studies.

Other dysregulated pathways include glutathione metabolism (Figure 6) and arginine and proline metabolism (Figure 7). Similar to the glutamate pathway, both are frequently reported for their relevance to ALS [25,64,65]. Studies have proposed that glutamate excitotoxity may cause depletion of glutathione (GSH), rendering neurons vulnerable to oxidative stress since GSH combats radical oxygen species. While evidence points to oxidative stress as part of the mechanism of progression in ALS, this connection is not well understood [62]. While less discussion in the field of ALS concentrates on arginine and proline metabolism [54], creatine and creatinine metabolism are included within this pathway, and the ratio of these analytes has been utilized to discriminate ALS patients from healthy individuals [60,66].

**Figure 6 metabolites-12-01096-f006:**
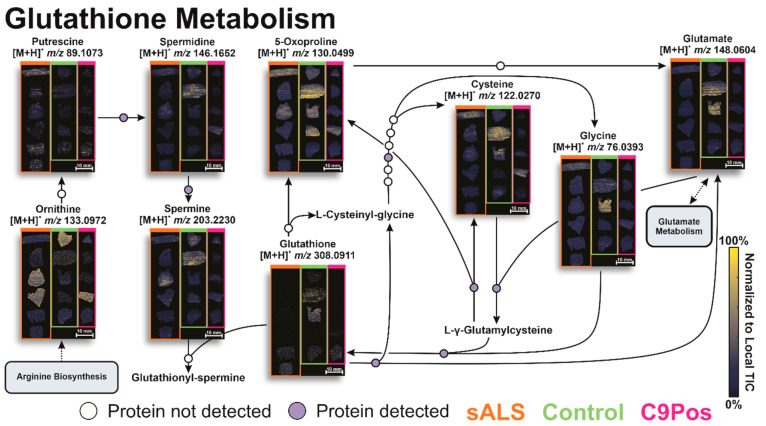
Annotated glutathione metabolism pathway with associated metabolite ion images by IR-MALDESI. Ion images are separated by disease classification respectively with orange, green, and pink boxes (sALS, control, C9Pos). Metabolomic conversions are indicated by arrows with white and/or purple circles based on protein detection. The number of circles with each arrow are indicative of the number of proteins involved in metabolite conversion as reported by KEGG database.

**Figure 7 metabolites-12-01096-f007:**
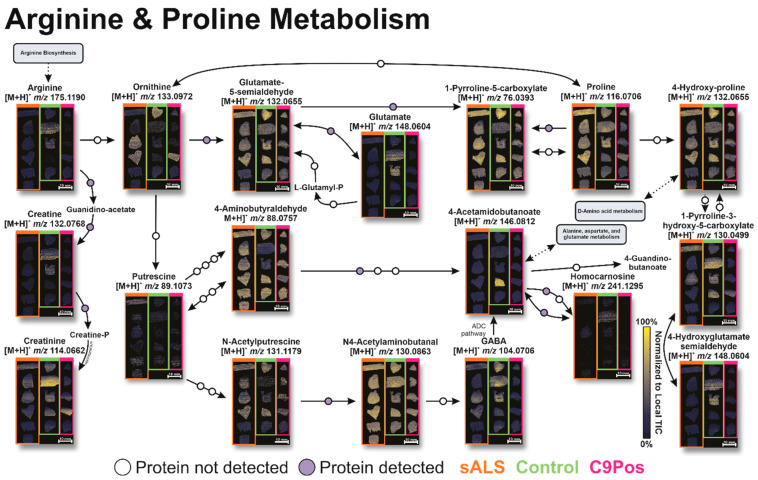
Arginine and proline metabolism pathway with associated metabolite ion mages by IR-MALDESI. Metabolite conversions with detected proteins are shown with ion images and purple or white circles along conversion arrows. The intensity of ion images indicates higher or lower abundance of metabolite detection. Circles along the pathway highlight detection or absence of proteins in the same patients. Disease cohorts for each ion are separated by color in each ion image (orange, green, and pink for sALS, Control, and C9Pos, respectively).

Consistent with the alanine, aspartate, and glutamate metabolism (Figure 5), no cohort demonstrated significant dysregulation throughout either pathway in Figure 6 and Figure 7. Some metabolites clearly show higher relative abundance and more homogeneous spatial distribution, such as proline (*m*/*z* 116.0706) and N4-Acetylaminobutanal (*m*/*z* 130.0863), but this did not evidently propagate in downstream conversions along the pathway for one group exclusively.

Deeper integration of proteomic findings is demonstrated in Figure 8, where tryptamine was converted to indole-3-acetaldehyde in part by a redox reaction of the CH-NH2 group by amine oxidase B (MOAB) enzyme [67]. Metabolite ion images from IR-MALDESI MSI analyses are coupled with box plots, which are based on the average ion abundance across each sample in each group.

Each group of boxplots is accompanied by a *p*-value from a one-way analysis of variance (ANOVA) to evaluate the effect of cohort identity on the detection of each metabolite. For example, Figure 8 reports a *p* = 0.075 with regard to the detection of tryptamine between disease classifications. Results for neither metabolite in Figure 8 were statistically significant by ANOVA tests. However, the average ion abundances of both tryptamine (${\bar{x}}_{t}$) and its converted metabolite, indole-3-acetaldehyde (${\bar{x}}_{i}$), follow the same descending order of sALS (${\bar{x}}_{t}$ = ~2.0 × 10^4^ ions/s, ${\bar{x}}_{i}$ = ~1.6 × 10^4^ ions/s), C9Pos (${\bar{x}}_{t}$ = ~1.3 × 10^4^ ions/s, ${\bar{x}}_{i}$ = ~1.1 × 10^4^ ions/s), and control (${\bar{x}}_{t}$ = ~1.1 × 10^4^ ions/s, ${\bar{x}}_{i}$ = ~9.2 × 10^3^ ions/s) cohorts. Congruent with lack of differences between cohorts in MSI data, the concentration of MOAB detected is similar between the groups (Eigenprotein Value = ~29.25) [47]. This example demonstrates the capability of deep multiomic integration with specific metabolite abundance and protein quantification for the purposes of studying ALS. Three other examples of enzymatic conversions can be found in Appendix A. The spectral accuracy of the converted metabolites is shown in Appendix A.

Ultimately, the metabolomic and multiomic results presented here parallel other reports on ALS. In combination with untargeted metabolite identification, IR-MALDESI MSI provided pivotal insight to analyte spatial distribution and the ability to integrate with protein detection. While providing this unique perspective, MSI experiments can be time-consuming when imaging large tissues and a large pool of samples. This study analyzed a small quantity of subjects of limited diversity (e.g., race, sex) at a single time point in one location of the brain. Follow-up studies could be made to increase the sample size and/or involve multiple brain locations while balancing the practical limitations, sample access/availability, and time. Additionally, including adequate patient samples from diverse racial backgrounds and both sexes would present more comprehensive and accurate conclusions in future studies. Nevertheless, this study helps to establish a foundation for future studies of ALS in MSI experiments, which can be further enhanced by multiomic integration for a wider perspective and understanding of the disease.

## 4. Conclusions

Ultimately, complementary results indicate subtle differences between sALS and *C9orf72* positive cohorts against control patients, proteomic and metabolomic alike. By IR-MALDESI MSI analyses, nearly 300 metabolites were putatively identified with HRAM-MS and compared across disease cohorts to evaluate differential expression, if present. Several metabolic pathways were identified with potential roles in the pathology of ALS based on differences in detection between groups, including alanine, aspartate, and glutamate metabolism, glutathione metabolism, and arginine and proline metabolism. However, when comparing detection of metabolites downstream, no obvious trends of up- or down-regulation were detected specific to one or more cohorts in the study indicative of a disease state. Further, this investigation provides an approach for combining LC-MS/MS proteomic data with MSI results for multiomic metabolic pathway enrichment with interpretation specific to ALS. Future endeavors will explore further and validate targeted pathways with more patients of diverse backgrounds, and potentially other sample types involving other genetic mutations, to understand more about the metabolic signature of ALS.

## Figures and Tables

**Figure 1 metabolites-12-01096-f001:**
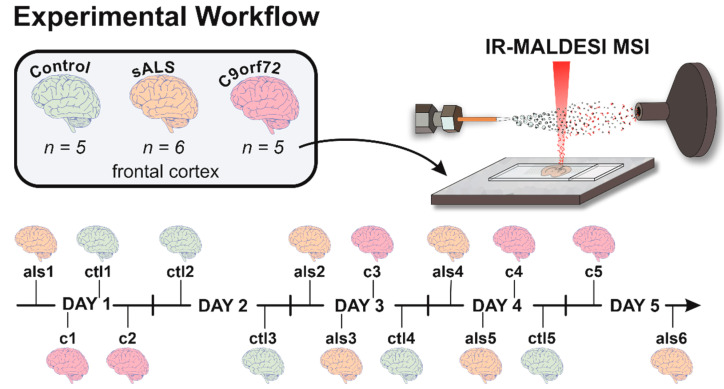
Summary of IR-MALDESI experimental workflow. Samples were sectioned and preserved at −80 °C prior to analysis without any other preparation. Using a randomized fashion, individual samples were run immediately after an energy-absorbing ice matrix was applied across the sample. Sample IDs were assigned after run-order was determined and replaced patient autopsy numbers.

**Figure 2 metabolites-12-01096-f002:**
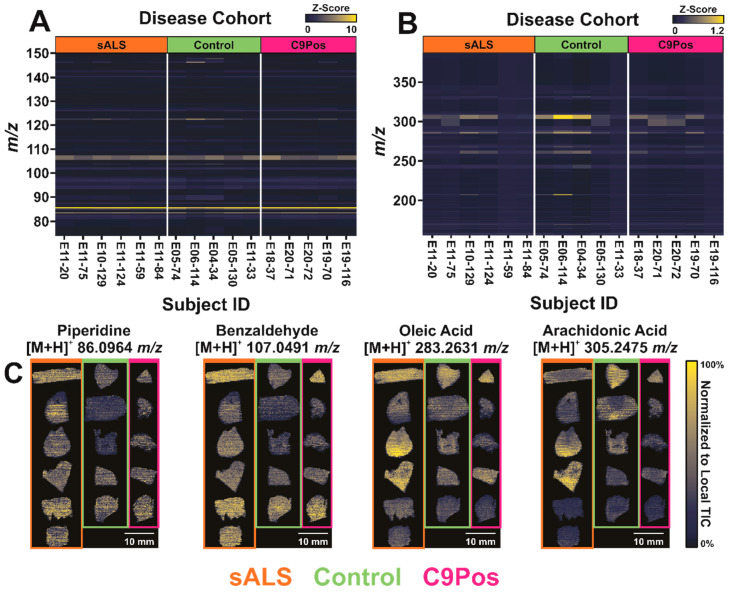
Heat maps illustrating the abundance of detected metabolites categorized by disease classification with examples of putative identifications. All abundance values were *z*-transformed to highlight differences between groups, where yellow and blue respectively show higher or lower abundance values relative to the individual samples’ mean and standard deviation. Samples are separated by both disease category and subject identification (e.g., als1, ctl1, and c1 for sALS, control, and C9Pos patients, respectively). (**A**,**B**) respectively isolate *m*/*z* 75–150 and *m*/*z* 150–400. (**C**) Putative identifications with ion images for all samples are shown in conjunction with annotations from heat maps.

**Figure 3 metabolites-12-01096-f003:**
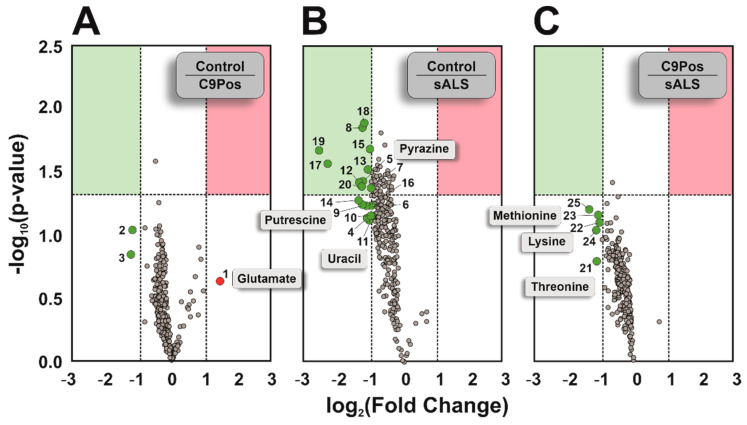
Volcano plots comparing the various disease cohorts. The highlighted regions (green and red) indicate metabolites with significant *p*-values (α = 0.05) from a Student’s *t*-test beyond a two-fold change, where the ratio for the calculated fold change is indicated in each plot. Seven putative identifications are indicated (e.g., glutamate, methionine). All numbered annotations correlate to metabolite annotations in Appendix A.

**Figure 4 metabolites-12-01096-f004:**
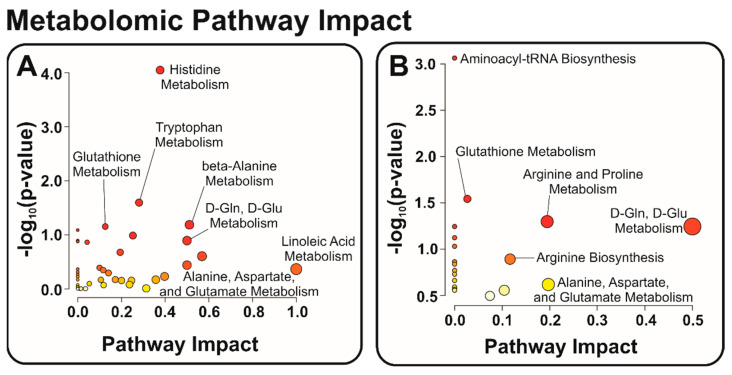
Annotated pathway impact graphs created in MetaboAnalyst 5.0. Data points vary in size and color depending on impact and significance to biological pathways. Panel (**A**) shows relevant pathways for all metabolites annotated in the study, whereas panel (**B**) isolates pathways related to the significant metabolites as indicated by the volcano plots in Figure 3 (reported in Appendix A).

**Figure 8 metabolites-12-01096-f008:**
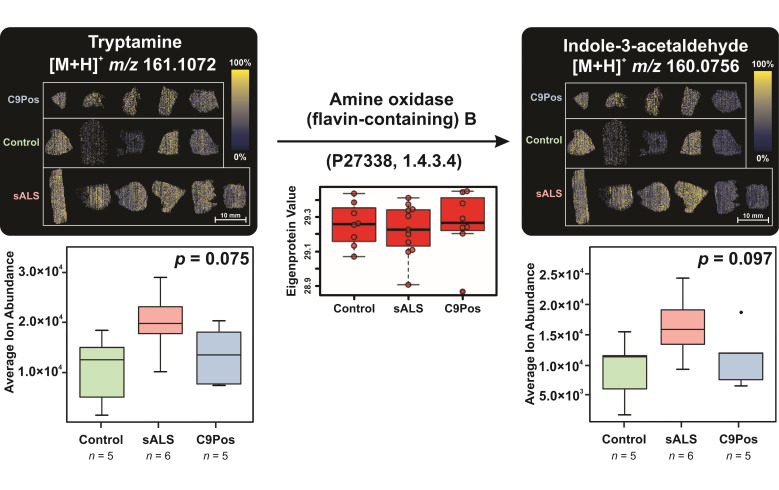
Metabolomic conversion of tryptamine to indole-3-acetaldehyde by amine oxidase B, highlighting multiomic integration with metabolomic and previously collected proteomic data. Ion images represent metabolomic detection by IR-MALDESI from post-mortem brain samples separated by disease classification, C9Pos, control, and sALS (blue, green, pink). Ion images are accompanied with boxplots of average on-tissue ion abundance for each sample and were tested for significance using a one-way ANOVA with *p*-values reported with their respective plots. The protein involved in conversion, amine oxidase B (MOAB), is shown with its Uniprot accession ID and EC number. Other examples of metabolomic conversion can be found in Appendix A, with confirmed spectral accuracy in Appendix A.

## Data Availability

All IR-MALDESI MSI datasets are available publicly in METASPACE: https://metaspace2020.eu/project/Sohn_ALS-Metabolomics_2022.

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
