# Peer review of "Interrogating the Metabolomic Profile of Amyotrophic Lateral Sclerosis in the Post-Mortem Human Brain by Infrared Matrix-Assisted Laser Desorption Electrospray Ionization (IR-MALDESI) Mass Spectrometry Imaging (MSI)"

_metabolites, 2022, doi:10.3390/metabo12111096_

Round 1
Reviewer 1 Report
Article review Interrogating the Metabolomic Profile of Amyotrophic Lateral 2 Sclerosis in Post-Mortem Human Brain by Infrared Matrix-As- 3 sisted Laser Desorption Electrospray Ionization (IR-MALDESI) 4 Mass Spectrometry Imaging (MSI) Work entitled "Interrogating the Metabolomic Profile of Amyotrophic Lateral 2 Sclerosis in Post-Mortem Human Brain by Infrared Matrix-As- 3 sisted Laser Desorption Electrospray Ionization (IR-MALDESI) 4 Mass Spectrometry Imaging (MSI) "is interesting, but the results are presented in a disorderly fashion, making it chaotic Results and Discussions 1. the work does not have the layout recommended by the editorial office (https://www.mdpi.com/journal/metabolites/instructions) 2. no title for figures 3. the material and methodology section lacked information on the subject of the studied group of patients. no information on the statistical methods used 4. There is no discussion section in which the obtained research results will be discussed and confronted with the current state of knowledge on the topic in question 5. conclusions must be legible and constitute a response to the stated aims of the work
Author Response
The authors deeply appreciated the time and comments provided by the reviewers to improve the quality and scientific contributions of this manuscript. All the comments are bolded with detailed responses below from the authors. Significant changes were made to the original manuscript and are highlighted with the Track Changes tool.
Referee 1:
Article review Interrogating the Metabolomic Profile of Amyotrophic Lateral Sclerosis in Post-Mortem Human Brain by Infrared Matrix-Assisted Laser Desorption Electrospray Ionization (IR-MALDESI) Mass Spectrometry Imaging (MSI) Work entitled "Interrogating the Metabolomic Profile of Amyotrophic Lateral Sclerosis in Post-Mortem Human Brain by Infrared Matrix-Assisted Laser Desorption Electrospray Ionization (IR-MALDESI) Mass Spectrometry Imaging (MSI)” is interesting, but the results are presented in a disorderly fashion, making it chaotic.
We apologize for the confusing organization in the initial submission and fully agree with the concerns mentioned. We have reformatted the manuscript accordingly. Thank you for your time and recommendations.
Results and Discussions
- The work does not have the layout recommended by the editorial office (https://www.mdpi.com/journal/metabolites/instructions).
We appreciate your attention to detail and have modified the manuscript to follow the recommended manuscript order as described in the “Instructions for Authors” webpage (https://www.mdpi.com/journal/metabolites/instructions). The revised version of the manuscript contains the research sections as follows: Introduction (p. 1-2), Materials and Methods (p. 2-4), Results and Discussions (p. 4-11), and Conclusions (p. 11). All references have been reordered appropriately.
- No title for figures.
Thank you for this suggestion. All figures, with the exception of Figures 2, 3, and 8, were modified to include appropriate titles. Figure 2, 3 and 8 contained different plot types as well as expressed different information, so it is not practical to encapsulate all the information into one simple title.
- The material and methodology section lacked information on the subject of the studied group of patients.
Further details have been added to the Materials and Methods to describe the subject groups analyzed and the number of subjects in each cohort. All clinical details for each patient that are allowed to be disclosed are included in the Supplemental Information (Table S1),including race, sex, age of ALS onset/death, diagnosis, and cause of death (where applicable). Additionally, based off of comment 2 from Reviewer 2, the cause of death for control patients has been disclosed and the subject IDs have been modified to protect the anonymity of patients in the study along with the respective figures. The Materials and Methods of the manuscript have been modified to read (p. 3, lines 103-109):
“All patients were diagnosed and cared for by JDG. Cases selected for this study were from the same cohort previously published for proteomic analysis [47]. Subject groups included control (n = 5), sporadic ALS (n = 6), and sporadic ALS with C9orf72 mutations (n = 5). C9orf72 mutations were confirmed previously in blood samples by a primed polymerase chain reaction (PCR) method (Umoh PMID 27488601). All patient case information is available in Table S1.”
- No information on the statistical methods used.
The details of statistical testing method have been added to the Materials and Methods in “3.4. Metabolomic Pathway Analysis and Proteomic Integration” (p. 4, lines 174-178). The information of statistical testing is also included in the respective figure captions (Figures 3, 8, S2), where Student’s t-tests were used for evaluating statistical significance, for emphasis and clarity.
“Statistical analyses were conducted for constructing the volcano plots in Figure 3 to compare statistical significance between patient cohorts of detected metabolites with Student’s t-tests, where a p-value (α) ≤ 0.05 was considered statistically significant. Additionally, metabolomic conversion figures (Figures 8 and S2) were coupled with p-values from Student’s t-tests to determine statistical significance between group metabolite abundances.”
- There is no discussion section in which the obtained research results will be discussed and confronted with the current state of knowledge on the topic in question.
Thanks for the Reviewer’s suggestion. For the preparation of this work, the Results and Discussion were combined and presented in tandem with one another for various reasons. Integration of MSI metabolomics results presented here and proteomic results from Umoh et al. [1] was a primary aim of this work. The authors think that combining the multiomic findings and biological interpretation with pathways would be easier for the reader when these sections are combined. Additionally, since this work is multiomic in nature, a reader may be from either metabolomics or proteomics, and we would like to combine the information here to demonstrate an example of multiomic integration and with the biological pathways discussed. Further, other works including Xi et al. [2] have presented work in a similar format that has been published in this journal. Therefore, this is the preferred format we would like to present this publication.
- Conclusions must be legible and constitute a response to the stated aims of the work.
To address the primary aims of the work, the conclusions have been modified to include a direct response to the results of the study (p. 11, lines 353-356). This work demonstrated: 1) the capability and an approach for multiomic MSI data integration specific to ALS, 2) detection of nearly 300 putatively-identified metabolites across 16 post-mortem human brain tissues to investigate their differential expression across disease cohorts, and 3) that proteomic and metabolomic analyses agreed in subtle differences between different ALS classifications presented here, but indicated several biological pathways that may be targeted for future studies.
“Ultimately, complementary results indicate subtle differences between cohorts, which is consistent with other studies investigating ALS. Nearly 300 metabolites were putatively identified and compared across disease cohorts and several metabolic pathways were identified with potential roles in the pathology of ALS, including alanine, aspartate, and glutamate metabolism, glutathione metabolism, and arginine and proline metabolism. Further, this investigation provides an approach for combining LC-MS/MS proteomic data with MSI results for multiomic metabolic pathway enrichment.”
- Umoh, M.E.; Dammer, E.B.; Dai, J.; Duong, D.M.; Lah, J.J.; Levey, A.I.; Gearing, M.; Glass, J.D.; Seyfried, N.T. A Proteomic Network Approach across the ALS‐FTD Disease Spectrum Resolves Clinical Phenotypes and Genetic Vulnerability in Human Brain. EMBO Mol. Med. 2018, 10, 48–62. https://doi.org/10.15252/emmm.201708202.
- Xi, Y.; Muddiman, D. C. Enhancing Metabolomic Coverage in Positive Ionization Mode Using Dicationic Reagents by Infrared Matrix-Assisted Laser Desorption Electrospray Ionization. Metabolites. 2021, 11, 810. https://doi.org/10.3390/METABO11120810/S1.
Reviewer 2 Report
In this manuscript, by Sohn et al., employs IR-MALDESI MSI to study dysregulated metabolites associated to ALS, aiming to identify biomarkers for precision diagnosis and therapeutic intervention for this fatal neurodegenerative disease. It addresses an important research subject and uses effective experimental methods. However, the sample size of the study is relatively low with racial diversity between groups. While some interesting results are presented, they are not conclusive. The manuscript may be improved by increasing case numbers or adding additional experiments/discussions to better integrate the IR-MALDESI MSI data with functional studies, proteomic analyses, or the previous work by Umoh et al., EMBO Mol Med. 2018 (Ref. # 47). As it stands now, the result of the study is not robust. It would also be helpful to define the condition of the normal controls who died in their 40-50’s.
Author Response
Referee 2:
In this manuscript, by Sohn et al., employs IR-MALDESI MSI to study dysregulated metabolites associated to ALS, aiming to identify biomarkers for precision diagnosis and therapeutic intervention for this fatal neurodegenerative disease. It addresses an important research subject and uses effective experimental methods.
The authors thank the Reviewer for their time and effort to review and provide comments to improve this manuscript.
- However, the sample size of the study is relatively low with racial diversity between groups. While some interesting results are presented, they are not conclusive. The manuscript may be improved by increasing case numbers or adding additional experiments/discussions to better integrate the IR-MALDESI MSI data with functional studies, proteomic analyses, or the previous work by Umoh et al., EMBO Mol Med. 2018 (Ref. # 47). As it stands now, the result of the study is not robust.
Thank for the Reviewer’s concerns with this, and the authors agree on the importance of racial and sex-related diversity in scientific studies. This aspect of the study could have been more expansive and inclusive to make more accurate conclusions on the variations between healthy and ALS-diagnosed individuals. While the authors can provide an explanation for the lack of diversity, we hope an addition to the manuscript (p. 11, lines 336-342 and 358-361) can help highlight the importance of inclusivity in other works to provide more accurate and widely applicable scientific conclusions.
Lines 336-342: “This study analyzed a small quantity of subjects of limited diversity (e.g., race, sex) at a single time point in one location of the brain. Follow-up studies could be made to increase the sample size and/or involve multiple brain locations while balancing the practical limitations, sample access/availability, and time. Additionally, including adequate patient samples of a diverse racial backgrounds and sex would present more comprehensive and accurate conclusions in future studies.”
Lines 358-361: “Future endeavors will explore further and validate targeted pathways with more patients of diverse backgrounds and potentially other sample types involving other genetic muta-tions to understand more about the metabolic signature of ALS.”
One of the primary goals of the work was to study and integrate the metabolomic MSI data collected with the proteomic information from previously completed LC-MS/MS experiments [1]. To accurately integrate these datasets, it was imperative to use the same patients between studies, thus limiting our case quantity and diversity by the patients Umoh et al. conducted analyses on.
Additionally, ALS is a rare disease that impacts a small population of people with limited specialty centers for treatment. Within this small frame of individuals were financial and geographical limitations of patients that were able to visit Emory University for ALS treatment, and then elected to donate samples to their repository. This creates practical challenges in sample acquisition for studies and ultimately limits the diversity of patients that may be utilized in a study such as this one.
In the future, the authors would expand the sample diversity and quantity if applicable for investigating more MSI and proteomics studies to draw more robust scientific conclusions. Again, one primary goal here was to offer a method to study and integrate the metabolomic MSI data with proteomics LC-MS/MS analysis.
- It would also be helpful to define the condition of the normal controls who died in their 40-50’s.
The authors fully agree with this perspective and think this information could contribute to the knowledge of the reader and any future studies. After confirming the ability to release this material, Table S1 has been modified to include the cause of death for all control patients in a separate column. All patient IDs in Table S1 and respective figures (Figures 1 and 2) have been modified from the true autopsy numbers to ambiguous sample IDs assigned based on order of analysis and their disease state. Control, sALS only, and C9orf72-confirmed individuals are now labelled as “ctrl,” “als,” and “c” with the run order number respectively. Figure 2 has also been revised from the initial submission to present heat maps in run order for each individual cohort.
References
- Umoh, M.E.; Dammer, E.B.; Dai, J.; Duong, D.M.; Lah, J.J.; Levey, A.I.; Gearing, M.; Glass, J.D.; Seyfried, N.T. A Proteomic Network Approach across the ALS‐FTD Disease Spectrum Resolves Clinical Phenotypes and Genetic Vulnerability in Human Brain. EMBO Mol. Med. 2018, 10, 48–62. https://doi.org/10.15252/emmm.201708202.
- Xi, Y.; Muddiman, D. C. Enhancing Metabolomic Coverage in Positive Ionization Mode Using Dicationic Reagents by Infrared Matrix-Assisted Laser Desorption Electrospray Ionization. Metabolites. 2021, 11, 810. https://doi.org/10.3390/METABO11120810/S1.
Round 2
Reviewer 1 Report
The introduced corrections significantly increased the value of the work, however, the article still does not have the layout recommended by the editors. Moreover, the conclusions are incorrectly constructed. Applications cannot repeat the admission. If we use abbreviations it is consistent throughout the work.
Author Response
The authors thank the Reviewers for their thorough comments and review of this manuscript. We appreciate your suggestions and have bolded all comments with detailed responses from the authors below. All revisions made to the initial manuscript submitted in round 1 are highlighted with the Track Changes tool. All page and line references in responses were included when the manuscript is showing “All Markup” from the “Review” tab.
Referee 1:
The authors thank you for your helpful feedback in both rounds of review. We appreciate your suggestions and feedback to improve the quality of this work.
- The introduced corrections significantly increased the value of the work, however, the article still does not have the layout recommended by the editors.
Thank you for your diligence in reviewing this article and considering the recommended format for this manuscript. The authors modified the original manuscript as advised to follow the listed sections as described (https://www.mdpi.com/journal/metabolites/instructions): Introduction (p. 1-2), Materials and Methods (p. 2-4), Results and Discussions (p. 4-11), and Conclusions (p. 11). All references were reordered accordingly to adjust for this change in formatting.
With regard to the Results and Discussion being presented simultaneously, the authors justify combining these sections and presenting them together in order to 1) enhance the readability for the audience to better comprehend the multiomic integration and biological interpretation, and 2) provide an example for other MSI multiomic workflows with biological pathways. Additionally, other published articles in Metabolites have followed a similar formatting pattern [1,2] in order to effectively communicate their results.
The authors understand that the separation of these two sections may be done in order to communicate the results and interpretation individually. However, for comprehensive discussion of our results and interpretation we believe conjoining these sections will enhance reader comprehension and the efficacy of the biological interpretation of the pathways presented. By including the findings reported here in conjunction with commentary of the current state of knowledge, the authors feel that this can further emphasize that the results parallel the findings in other studies and elucidate biological pathways that can be further investigated with regard to their pathogenesis and/or pathophysiology of ALS. The authors present this rationale as support for the preference in delivering the Results and Discussion together in this manuscript.
- Moreover, the conclusions are incorrectly constructed. Applications cannot repeat the admission.
The authors apologize for this oversight in the previous version of the manuscript. From this comment, it is the authors’ understanding that there are concerns with repeating the overarching context of the disease in literature (p. 14, line 479) rather than focusing on the results of the study and their proper communication in the Conclusions section. To remedy this concern, we offer modifications to the Conclusions section to focus on the results of the investigation presented in this work. This involved removing some elements from other studies that were mentioned (i.e., “…which is consistent with other studies investigating ALS” p. 14, line 479) and adding details of conclusions specific to what was discovered in this investigation (p. 14-15, lines 478-489), such as the quantity of putative identifications of metabolites and the lack of a clear trend (up- or down-regulated) within a particular cohort of the study.
- If we use abbreviations it is consistent throughout the work.
Thank you for your attention to this detail. The authors have reviewed this work and locations where a specific sample is labeled, the assigned Patient ID (e.g., als#, ctl#, c#) was utilized and consistently labeled. This labeling system is defined in “Section 2.1. Case Details and Sample Preparation” when introducing the cohorts of the study (p. 3, lines 105-107) where control, sALS, and C9Pos patients are denoted as “ctl”, “asl”, and “c” respectively. Figures 1, 2 and Table S1 were prepared to accurately reflect these abbreviations, as well as any specific patient references in the body of the article (e.g., p. 7, line 226). Where the whole patient cohort was referenced, the full condition labels (i.e., sALS, Control, C9Pos) were used for clarity in the appropriate figures or portions of the manuscript (e.g., p. 7, line 246). Such a labeling scheme was demonstrated in Figures 1-3, 5-8, and S1-2.
Further, the authors evaluated the body of the manuscript for any inconsistencies in other abbreviations beyond patient IDs and cohort distinctions, and all modifications required were made with Track Changes as necessary.
Reviewer 2 Report
Although some of my comments are not fully addressed, this might be due to limitations in resources.
Author Response
Referee 2:
Although some of my comments are not fully addressed, this might be due to limitations in resources.
The authors are grateful for your critical feedback to make this study more diverse and comprehensive. While we regret we could not expand further to make the patient cohorts more inclusive for more robust conclusions, we hope to emphasize the necessity of including a variety of patients where investigators are able to fully encapsulate and study metabolomic changes with regard to ALS to accurately reflect the entire population.
Round 3
Reviewer 1 Report
Thank you for the changes, but the conclusions still need to be corrected
472-475 – delete the first three sentences of the sentence in the conclusions. Conclusions must be specific and refer only to the research presented
Author Response
Thank you for your feedback and clarity of the previous suggestion. The authors have modified the manuscript as requested and the first three sentences of the Conclusions have been removed from the manuscript. The Conclusions previously read (previously, p. 14, lines 472-478):
“Amyotrophic lateral sclerosis is an insidious neurodegenerative disease…were conducted. Ultimately, complementary results indicate subtle differences between cohorts, proteomic and metabolomic alike…”
After removing the specified sentences and other modifications, the beginning of the Conclusions now read (currently, p. 14, lines 477-480):
“Ultimately, complementary results indicate subtle differences between sALS and C9orf72 positive cohorts against control patients, proteomic and metabolomic alike. By IR-MALDESI MSI analyses, nearly 300 metabolites were putatively identified with HRAM-MS analyses…”
Thank you for your feedback to correct the conclusions to be more appropriate for this manuscript. We appreciate your time and suggestions during the review of this manuscript.